# Adapting Neural Architectures Between Domains

**Yanxi Li** [1], **Zhaohui Yang** [2,3], **Yunhe Wang** [2], **Chang Xu** [1]

[1] School of Computer Science, University of Sydney, Australia
[2] Noah's Ark Lab, Huawei Technologies, China
[3] Key Lab of Machine Perception (MOE), Department of Machine Intelligence,
Peking University, China
`yali0722@uni.sydney.edu.au`, `zhaohuiyang@pku.edu.cn`,
`yunhe.wang@huawei.com`, `c.xu@sydney.edu.au`

## Abstract

Neural architecture search (NAS) has demonstrated impressive performance in automatically designing high-performance neural networks. The power of deep neural networks is to be unleashed for analyzing a large volume of data (e.g. ImageNet), but the architecture search is often executed on another smaller dataset (e.g. CIFAR-10) to finish it in a feasible time. However, it is hard to guarantee that the optimal architecture derived on the proxy task could maintain its advantages on another more challenging dataset. This paper aims to improve the generalization of neural architectures via domain adaptation. We analyze the generalization bounds of the derived architecture and suggest its close relations with the validation error and the data distribution distance on both domains. These theoretical analyses lead to AdaptNAS, a novel and principled approach to adapt neural architectures between domains in NAS. Our experimental evaluation shows that only a small part of ImageNet will be sufficient for AdaptNAS to extend its architecture success to the entire ImageNet and outperform state-of-the-art comparison algorithms.

## 1 Introduction

Neural architecture search (NAS) is to automate the design of neural architectures for networks. Recently, convolutional neural networks (CNNs) designed by NAS methods have already reached better performance than those manually designed ones on ImageNet. However, early NAS methods are computationally intensive, because of their demand for training and evaluation of a large number of architectures [20, 26]. This, therefore, makes it intractable to directly conduct the architecture search on large-scale benchmarks like ImageNet. As a trade-off, many NAS methods search on proxy tasks, such as CIFAR-10, and then retrain obtained architectures on ImageNet. However, even though on CIFAR-10, it is common for most methods to cost thousands of GPU days.

In the past years, great efforts were undertaken to significantly reduce the architecture search cost. Remarkably, the line of research on the differentiable manner for architecture search [17] have reduced the search cost dramatically to several GPU days or several GPU hours on the CIFAR-10 dataset. DARTS [17] relaxes discrete NAS search space as continuous architecture parameters and constructs a super network by weighted mixing of all candidate operations. In the super network, network weights and architecture parameters can be jointly optimized with gradient descent. The search cost of DARTS on CIFAR-10 is only 1 GPU day, but the parallel optimization of all candidate operations demands large GPU memory. GDAS [7] tackles this issue by using a differentiable sampler and sampling one operation per connection in each epoch. This dramatically reduces the usage of GPU memory, and the search can be completed within about 4 to 5 GPU hours depending on the setting. As GDAS reduces the width of the super network, P-DARTS [6] starts with a shallow super network and progressively increases its depth. This method slightly increases the search cost to about 7 GPU hours but can reach a better test performance. Besides direct reducing the training cost, CARS [24] proposes a novel efficient continuous evolutionary approach based on the historical evaluation.

Similarly, PVLL-NAS [16] schedules their evaluation with a performance estimator, who samples neural architectures for both architecture searching and iterative training of the estimator itself.

However, due to the inconsistent performance of architectures on different domains, searching on proxy tasks and then reusing the obtained architectures on large-scale benchmarks has become a rut, which may result in a huge generalization gap of the neural architecture on the two different domains. This generalization gap could be either positive or negative, but reflecting in practice, it would cause either omitting of good architectures or choosing of poor architectures and make the performance on the desired domain uncontrollable.

A few attempts are trying to break out of the rut by directly executing the architecture search on ImageNet. MnasNet [21] was established within the framework of reinforcement learning and costed 288 TPU days for one search on ImageNet. By applying the differentiable NAS techniques, ProxylessNAS [5] binarized architectures to boost search speed and reduced the search cost on ImageNet to 8.33 GPU day, but it is still about 28 to 52 times slower than its counterparts [7, 6, 23, 25] on CIFAR-10. NAS on CIFAR-10 is fast but deploying the searched architecture on ImageNet will receive the accuracy fluctuation; directly searching on ImageNet is slow but its architecture accuracy can be guaranteed. If it is infeasible to swallow the entire ImageNet, we ask whether a smaller part of ImageNet on top of the efficient NAS on CIFAR-10 would rescue us from this dilemma.

In this paper, we consider the inconsistency in the generalization of architectures from a new perspective of adapting neural architectures between domains. The proxy task such as CIFAR-10 for searching is considered as the source domain, and the large-scale benchmark such as ImageNet to deploy searched architectures for testing or application is our aiming target domain. Firstly, the relationship between the empirical source validation error and the expected target error of neural architectures is analyzed. Since NAS approaches typically optimize network weights during the training phase and then search for architectures during the validation phase, it is meaningful to find a generalization bound by validation. Two versions of the generalization bound are proposed. One associates with the source validation error, while another introduces an additional target validation error calculated on a subset of target samples. Based on them, we propose a lightweight method to explicitly minimize the cross-domain generalization gap of neural architectures during NAS. We name it as **Adapt**able **N**eural **A**rchitecture **S**earch (**AdaptNAS**). The generalisability and efficiency of AdaptNAS are demonstrated with extensive experiments. On the three digits dataset, we show that AdaptNAS generalizes better than baselines without generalization constraint. Then, large-scale experiments are performed on CIFAR-10 and ImageNet and compared with different state-of-the-art NAS methods that search either with proxy tasks or directly on ImageNet.

## 2 Related Work

Existing NAS methods with proxy typically use CIFAR-10 as a proxy task and directly generalize their obtained architectures to ImageNet without any constraint. Early methods [20, 26] can easily cost thousands of GPU days to find an architecture even on CIFAR-10. The emerging of differentiable search methods, represented by DARTS [17], reduces search cost to one or several GPU days. A variant of DARTS, GDAS [7], further reduces the search cost to several GPU hours by proposing a differentiable architecture sampler. In terms of selection of the proxy task, P-DARTS [6] uses CIFAR-100 as one of their proxy tasks. CIFAR-100 contains fine-grained categories but identical sample size and number comparing to CIFAR-10 [14]. Thus, the search cost of P-DARTS on CIFAR-10 and CIFAR-100 are similar (0.3 GPU days). FBNet [22] and HM-NAS [23] searches on a subset of ImageNet with 100 classes rather than the entire 1000 classes. The 100-class ImageNet can be considered as a new proxy task. ProxylessNAS [5] can directly search on the ImageNet to avoid the gap in generalization, but the search cost increases to 8.33 GPU days. MdeNAS [25] also directly searches on Imagenet, but they boost search speed by search with the MobileNetV2 [11] as a backbone and reuse the structure found in [5] instead of search from scratch, which limits it potential.

## 3 Generalization Analysis for AdaptNAS

Let $\mathcal{X}$ be the input data space, $\mathcal{Z}$ be a latent representation space and $\mathcal{Y}$ be the label space. A convolutional neural network (CNN) $f : \mathcal{X} \rightarrow \mathcal{Y}$ can be disassembled into a representation mapping $\mathcal{R} : \mathcal{X} \rightarrow \mathcal{Z}$ and a classification hypothesis $h : \mathcal{Z} \rightarrow \mathcal{Y}$. In general, $h$ is usually a naive single layer feed-forward network with weights $\boldsymbol{w}_h$, and $\mathcal{R}$ can have complex topology described by network weights $\boldsymbol{w}_{\mathcal{R}}$ and the neural architecture $\boldsymbol{A}$. The target of NAS is to find an optimum architecture

$\boldsymbol{A}^* \in \mathbb{A}$ that minimize the classification loss $\mathcal{L}(\boldsymbol{w}^*(\boldsymbol{A}), \boldsymbol{A}) = \mathbb{E}_{\boldsymbol{x}_i \sim \mathcal{D}}[\ell(f(\boldsymbol{x}_i; \boldsymbol{w}^*(\boldsymbol{A}), \boldsymbol{A}), y_i)]$:

$$\boldsymbol{A}^* = \arg\min_{\boldsymbol{A}} \mathcal{L}(\boldsymbol{w}^*(\boldsymbol{A}), \boldsymbol{A}), \tag{1}$$

where $\mathbb{A}$ is a predefined search space, $\mathcal{D}$ is a distribution over the input space $\mathcal{X}$, $\ell$ is a loss function and $\boldsymbol{w}^*(\boldsymbol{A})$ is the optimal value of network weights $\boldsymbol{w} = \{\boldsymbol{w}_{\mathcal{R}}, \boldsymbol{w}_h\}$ depending on the current architecture $\boldsymbol{A}$. We consider the bi-level optimization form of NAS:

$$\min_{\boldsymbol{A}} \quad \mathcal{L}_{valid}(\boldsymbol{w}^*(\boldsymbol{A}), \boldsymbol{A}) \tag{2}$$

$$\textbf{s.t.} \quad \boldsymbol{w}^*(\boldsymbol{A}) = \arg\min_{\boldsymbol{w}} \mathcal{L}_{train}(\boldsymbol{w}, \boldsymbol{A}), \tag{3}$$

where $\mathcal{L}_{train}$ and $\mathcal{L}_{valid}$ are losses on the training distribution $\mathcal{D}_{train}$ and the held-out validation distribution $\mathcal{D}_{valid}$, respectively. In such a bi-level form, $\boldsymbol{w}$ and $\boldsymbol{A}$ are optimized alternately with Eqs. (3) and (2) until convergence or reach a maximum iteration number.

We define *ProxyNAS* as those existing NAS methods that conduct optimization on a relatively small proxy task (e.g. CIFAR10) and evaluate the searched architectures on the large-scale task (e.g. ImageNet). In this paper, we tend to revisit such a tradition of NAS training and evaluation from the perspective of domain adaptation and propose the AdaptNAS. The smaller training data of the architecture is taken as *source domain*, and we can also leverage a few data from the *target domain* (e.g. ImageNet) to improve the generalization of the architecture.

Formally, a domain can be considered as a pair of a distribution $\mathcal{D}$ on input space $\mathcal{X}$ and a labeling function $f : \mathcal{X} \to \mathcal{Y}$. We can thus define the source and target domains as $\langle \mathcal{D}_S, f_S \rangle$ and $\langle \mathcal{D}_T, f_T \rangle$, respectively. In this section, we first introduce a generalization bound in NAS constrained by the source domain validation error and a domain distance. Then, we introduce the target domain validation error into the boundary to utilize any accessible target domain information. Detailed proofs are provided in the supplementary material.

### 3.1 Generalization Bounds via Validation of Source Domain

To quantify the generalization gap between domains, a domain distance measurement is necessary. We use the $\mathcal{A}$-distance [13] as the measurement. The $\mathcal{A}$-distance is defined as follow:

**Definition 1** ($\mathcal{A}$-distance). Let $\mathcal{D}$ and $\mathcal{D}'$ be distributions on $\mathcal{X}$, and $\mathcal{A}$ be a collection of subsets of $\mathcal{X}$ such that every $A \in \mathcal{A}$ is measurable w.r.t $\mathcal{D}$ and $\mathcal{D}'$. The $\mathcal{A}$-distance between $\mathcal{D}$ and $\mathcal{D}'$ is

$$d_{\mathcal{A}}(\mathcal{D}, \mathcal{D}') := 2 \sup_{A \in \mathcal{A}} |\Pr_{\mathcal{D}}[A] - \Pr_{\mathcal{D}'}[A]|, \tag{4}$$

where $\Pr_{\mathcal{D}}[A]$ is the probability of $A$ under $\mathcal{D}$.

The complexity of the $\mathcal{A}$-distance can be limited by the *symmetric difference hypothesis space* $\mathcal{H}\Delta\mathcal{H}$ [4]. For simplification, we discuss the binary classification scenario, where $\mathcal{Y} = \{0, 1\}$. The theory results can be easily generalized to the multi-class case. Under the binary setting, we have $\mathcal{H}\Delta\mathcal{H} = \{h(\boldsymbol{z}) \oplus h'(\boldsymbol{z})|h, h' \in \mathcal{H}\}$, where $\oplus$ is the XOR operation, and $\mathcal{H}$ is a hypothesis space. Based on this, $\mathcal{A}_{\mathcal{H}\Delta\mathcal{H}}$ can be defined as a collection of all subsets $A$ such that $A = \{\boldsymbol{x}|\boldsymbol{x} \in \mathcal{X}, h(x) \neq h'(x)\}$ for some $h, h' \in \mathcal{H}$. Letting $\mathcal{A} = \mathcal{A}_{\mathcal{H}\Delta\mathcal{H}}$ in Eq. 4, we can have the symmetric difference $\mathcal{A}$-distance, notated as $d_{\mathcal{H}\Delta\mathcal{H}}(\mathcal{D}, \mathcal{D}')$. The advantage of using $d_{\mathcal{H}\Delta\mathcal{H}}(\cdot, \cdot)$ is that it satisfies:

$$\forall h, h' \in \mathcal{H}, \quad |\varepsilon_S(h, h') - \varepsilon_T(h, h')| \leq \frac{1}{2} d_{\mathcal{H}\Delta\mathcal{H}}(\mathcal{D}_S, \mathcal{D}_T), \tag{5}$$

where $\varepsilon(\cdot, \cdot)$ measures the disagreement of two hypothesis. The measure in the source domain is defined as $\varepsilon_S(h, h') = \mathbb{E}_{\boldsymbol{x} \sim \mathcal{D}_S}[|h(\boldsymbol{x}) - h'(\boldsymbol{x})|]$, and we use a similar definition for the target domain. Similar to $\mathcal{D}_S, \mathcal{D}_T$, we notate the source and target latent distribution on $\mathcal{Z}$ as $\widetilde{\mathcal{D}}_S$ and $\widetilde{\mathcal{D}}_T$. The labelling functions from $\mathcal{Z}$ to $\mathcal{X}$ are represented by $\tilde{f}_S$ and $\tilde{f}_T$, respectively. We define the expected error of $h$ in a domain $S$ as the disagreement between $h$ and $\tilde{f}_S$, notated as $\varepsilon_S(h) := \varepsilon_S(h, \tilde{f}_S)$. The similar notation is also used for the target domain. Then, Eq. 5 can lead to Lemma 1.

**Lemma 1.** [4] *Let $\mathcal{R}$ be a representation function $\mathcal{R} : \mathcal{X} \to \mathcal{Z}$, and $\widetilde{\mathcal{D}}_S$ and $\widetilde{\mathcal{D}}_T$ be the source and target distribution over $\mathcal{Z}$, respectively. For $h \in \mathcal{H}$:*

$$\varepsilon_T(h) \leq \varepsilon_S(h) + \frac{1}{2} d_{\mathcal{H}\Delta\mathcal{H}}(\widetilde{\mathcal{D}}_S, \widetilde{\mathcal{D}}_T) + \lambda, \tag{6}$$

*where $\lambda$ is combined error of the optimum hypothesis $h^* = \arg\min_{h \in \mathcal{H}} \varepsilon_S(h) + \varepsilon_T(h)$ on both domains: $\lambda = \varepsilon_S(h^*) + \varepsilon_T(h^*)$.*

Lemma 1 reveals that the cross-domain generalization gap is bounded by the expected source error and the $\mathcal{A}$-distance of latent distributions. This distance can be minimized by optimizing the representation function $\mathcal{R}$. In NAS, $\boldsymbol{w} = \{\boldsymbol{w}_{\mathcal{R}}, \boldsymbol{w}_h\}$ are optimized over the training data, while in the validation phase, given the fixed $\boldsymbol{w}$, the architecture $\boldsymbol{A}$ is further optimized to minimize the validation error (see Eq. 2). We therefore proceed to extend the above analysis to the validation set. Let $\widetilde{\mathcal{U}}_{S,train}$ and $\widetilde{\mathcal{U}}_{S,valid}$ be a training set and a held-out validation set of i.i.d. sample drawn from $\widehat{\mathcal{D}}_S$, respectively, such that $\widetilde{\mathcal{U}}_{S,train} \cap \widetilde{\mathcal{U}}_{S,valid} = \varnothing$. The validation is of a subset $\mathcal{H}'$ of $\mathcal{H}$ depending on $\widetilde{\mathcal{U}}_{S,train}$ but is independent of $\widetilde{\mathcal{U}}_{S,valid}$. The following theorem provides an analysis on the expected target error in terms of the empirical source validation error on $\widetilde{\mathcal{U}}_{S,valid}$ and an empirical $\mathcal{A}$-distance.

**Theorem 2.** *Let $m$ be the size of $\widetilde{\mathcal{U}}_{S,valid}$, $d'$ be the VC-dimension of $\mathcal{H}'$, and $\widetilde{\mathcal{U}}_S$ and $\widetilde{\mathcal{U}}_T$ be sets of unlabelled i.i.d. samples drawn from $\widetilde{\mathcal{D}}_S$ and $\widetilde{\mathcal{D}}_T$, each with size $m'$. With probability at least $1 - \delta$, for $h \in \mathcal{H}'$:*

$$
\begin{aligned}
\varepsilon_T(h) \leq \hat{\varepsilon}_{S,valid}(h) &+ \frac{d' \log m - \log \delta}{3m} + \sqrt{\frac{2(d' \log m - \log \delta)}{m}} \\
&+ \frac{1}{2} d_{\mathcal{H} \Delta \mathcal{H}}(\widetilde{\mathcal{U}}_S, \widetilde{\mathcal{U}}_T) + 4\sqrt{\frac{d' \log(2m') + \log(4/\delta)}{m'}} + \lambda.
\end{aligned}
\tag{7}
$$

Theorem 2 provides an empirical estimate of the cross-domain generalizability of architectures by validation. The target expected error of an architecture $\boldsymbol{A}$ depends on two terms, the validation error of the entire network (including both $\mathcal{R}$ and $h$, but $h$ is fixed during the validation) in source domain and the $\mathcal{A}$-distance of $\widetilde{\mathcal{U}}_S$ and $\widetilde{\mathcal{U}}_T$ generated by the neural architecture.

### 3.2 Generalization Bounds via a Hybrid Validation

In Theorem 2, $\widetilde{\mathcal{U}}_{S,valid}$ is requested to compute $\hat{\varepsilon}_{S,valid}(h)$. Besides the validation set on the source domain, we could further have labeled samples from the target domain for the validation use in practice. A hybrid validation set of $m$ examples is therefore defined as the composition of $\beta m$ source examples and $(1 - \beta)m$ target examples, where $\beta \in [0, 1]$. Validation errors on the source and target domain are combined by weighted sum with $\alpha \in [0, 1]$:

$$
\hat{\varepsilon}_{\alpha,valid}(h) = \alpha \hat{\varepsilon}_{S,valid}(h) + (1 - \alpha)\hat{\varepsilon}_{T,valid}(h).
\tag{8}
$$

The following lemma bounds the expected target error with the expected hybrid error. This bound can be extended to the validation set as well.

**Lemma 3.** *Let $\varepsilon_\alpha(h)$ be an expected hybrid error weighted by $\alpha \in [0, 1]$. For $h \in \mathcal{H}$:*

$$
\varepsilon_T(h) \leq \varepsilon_\alpha(h) + \alpha \left( \frac{1}{2} d_{\mathcal{H} \Delta \mathcal{H}}(\widetilde{D}_S, \widetilde{D}_T) + \lambda \right).
\tag{9}
$$

By applying Lemma 3 to Theorem 2, we can have the following corollary.

**Corollary 4.** *Let $\alpha \in [0, 1]$ be the weight of the hybrid error, and $\beta \in [0, 1]$ be the ratio of i.i.d. samples drawn from $\widetilde{\mathcal{D}}_S$ and $\widetilde{\mathcal{D}}_T$ in a held-out validation set. With probability at least $1 - \delta$, for $h \in \mathcal{H}'$:*

$$
\begin{aligned}
\varepsilon_T(h) \leq \hat{\varepsilon}_{\alpha,valid}(h) &+ \left( \frac{\alpha}{\beta} + \frac{1 - \alpha}{1 - \beta} \right) \left( \frac{d' \log m - \log \delta}{3m} + \sqrt{\frac{2(d' \log m - \log \delta)}{m}} \right) \\
&+ \alpha \left( \frac{1}{2} d_{\mathcal{H} \Delta \mathcal{H}}(\widetilde{\mathcal{U}}_S, \widetilde{\mathcal{U}}_T) + 4\sqrt{\frac{d' \log(2m') + \log(4/\delta)}{m'}} + \lambda \right).
\end{aligned}
\tag{10}
$$

To utilize Corollary 4, $\alpha$ and $\beta$ need to be determined. When $\alpha = 1$ and the target validation error is not considered, Corollary 4 will be reduced to Theorem 2. With $\alpha \in (0, 1)$, we will introduce both source and target samples for validation, and the generalizability of architectures could be improved (see the $\alpha$ before $\mathcal{A}$-distance). The selection of $\beta$ is a trade-off. With a fixed source validation set, a smaller $\beta$ means more target samples and heavier computation cost. Besides, with a $\beta$ approaches 0 or 1, the source and target sample number becomes highly unbalanced and the factor $\alpha/\beta + (1 - \alpha)/(1 - \beta)$ approaches infinite, which makes the architecture optimization unpredictable. This therefore reminds us of carefully balancing the sample size in source and target domains. More empirical discussions can be found in experiments.

# 4 AdaptNAS Algorithm

Motivated by theorems in Section 3, we propose two versions of AdaptNAS. The former, **AdaptNAS-Source**, following Theorem 2, optimizes network weights with source training samples and estimates the $\mathcal{A}$-distance in the training phase. In the searching phase, the architecture $\boldsymbol{A}$ is optimized to reduce both the source validation loss and the $\mathcal{A}$-distance. The latter, **AdaptNAS-Combined**, following Corollary 4, uses a similar schema as AdaptNAS-S, but further considers a subset of target samples to optimize both network weights and architectures by utilizing Eq. 8.

It is intractable to directly compute the $\mathcal{A}$-distance, but we can approximate it with a domain discriminator [3]. With a domain discriminator $h_d \in \mathcal{H}$, we have:

$$d_{\mathcal{H}\Delta\mathcal{H}}(\widetilde{\mathcal{U}}_S, \widetilde{\mathcal{U}}_T) = 2\left(1 - 2\min_{h_d \in \mathcal{H}} \hat{\varepsilon}_d(h_d)\right), \tag{11}$$

where $\hat{\varepsilon}_d(h_d) = \frac{1}{2m'}\sum_{i=1}^{2m'} |h_d(\boldsymbol{z}_i) - y_{d,i}|$ is the empirical discrimination error on $\boldsymbol{z}_i \in \widetilde{\mathcal{U}}_S \cup \widetilde{\mathcal{U}}_T$, and $y_{d,i}$ is the domain label. Although the optimal $h_d$ is normally unsolvable, the $\mathcal{A}$-distance can still be approximated arbitrarily well by optimizing it. An useful property of Eq. 11 is that $d_{\mathcal{H}\Delta\mathcal{H}}(\widetilde{\mathcal{U}}_S, \widetilde{\mathcal{U}}_T) \propto \frac{1}{\min_{h_d \in \mathcal{H}} \hat{\varepsilon}_d(h_d)}$. With such an observation, it is possible to learn a domain discriminator during NAS and use adversarial learning to minimize the $\mathcal{A}$-distance by maximizing discrimination error. In Adapt-NAS, we first learn an $h_d$ to distinguish the latent representation produced by $\mathcal{R}$ in the training phase to minimize a discrimination loss: $\mathcal{L}_d(h_d; \boldsymbol{w}_{\mathcal{R}}, \boldsymbol{A}) = \mathbb{E}_{\boldsymbol{x}_i \sim \mathcal{D}_S \cup \mathcal{D}_T} [\ell(h_d(\mathcal{R}(\boldsymbol{x}_i; \boldsymbol{w}_{\mathcal{R}}, \boldsymbol{A})), d_i)]$. Then, $\boldsymbol{A}$ of $\mathcal{R}$ is optimized in the searching phase with an adversarial loss to maximize the discrimination loss.

In AdaptNAS-S, the lower-level optimization in Eq. 3 can be reformed as Eq. 13, where $\mathcal{L}_{S,train}(h; \boldsymbol{w}_{\mathcal{R}}, \boldsymbol{A}) = \mathbb{E}_{\boldsymbol{x}_i \sim \mathcal{D}_{S,train}} [\ell(h(\mathcal{R}(\boldsymbol{x}_i; \boldsymbol{w}_{\mathcal{R}}, \boldsymbol{A})), y_i)]$ is the source training loss. The similar notations are also used for the source validation loss and the target training and validation losses. Similarly, the upper-level optimization in Eq. 2 can be reformed as Eq. 12.

$$\min_{\boldsymbol{A}} \quad \mathcal{L}_{S,valid}(h; \boldsymbol{w}_{\mathcal{R}}, \boldsymbol{A}) - \mathcal{L}_d(h_d; \boldsymbol{w}_{\mathcal{R}}, \boldsymbol{A}), \tag{12}$$

$$\text{s.t.} \quad \max_{h_d} \min_{\boldsymbol{w}, h} \mathcal{L}_{S,train}(h; \boldsymbol{w}_{\mathcal{R}}, \boldsymbol{A}) - \mathcal{L}_d(h_d; \boldsymbol{w}_{\mathcal{R}}, \boldsymbol{A}). \tag{13}$$

However, the discriminator gradients at early stage could be noisy and will corrupt the entire network. To control them in back-propagation, we apply a gradient reversal technique [8]. In the training phase, with gradient reversal, $h$ and $h_d$ are still updated with their own gradients, but $\boldsymbol{w}_{\mathcal{R}}$ is updated with additional reversed discriminator gradients weighted by $\gamma$ as in Eq. 14. The weight term $\gamma \in [0, 1]$ can be dynamically adjusted during optimization. In the searching phase, by utilizing differentiable NAS [7, 17], architectures can be relaxed as continuous parameters and updated with adversarial learning as in Eq. 15.

$$\boldsymbol{w}_{\mathcal{R}} \leftarrow \boldsymbol{w}_{\mathcal{R}} - \eta \left(\frac{\partial \mathcal{L}_{S,train}(h; \boldsymbol{w}_{\mathcal{R}}, \boldsymbol{A})}{\partial \boldsymbol{w}_{\mathcal{R}}} - \gamma \frac{\partial \mathcal{L}_d(h_d; \boldsymbol{w}_{\mathcal{R}}, \boldsymbol{A})}{\partial \boldsymbol{w}_{\mathcal{R}}}\right), \tag{14}$$

$$\boldsymbol{A} \leftarrow \boldsymbol{A} \quad - \eta \left(\frac{\partial \mathcal{L}_{S,valid}(h; \boldsymbol{w}_{\mathcal{R}}, \boldsymbol{A})}{\partial \boldsymbol{A}} - \gamma \frac{\partial \mathcal{L}_d(h_d; \boldsymbol{w}_{\mathcal{R}}, \boldsymbol{A})}{\partial \boldsymbol{A}}\right). \tag{15}$$

In AdaptNAS-C, we cannot simply replace $\mathcal{L}_S(h; \boldsymbol{w}, \boldsymbol{A})$ by $\mathcal{L}_\alpha(h; \boldsymbol{w}, \boldsymbol{A})$, because Corollary 4 has already revealed that the $\mathcal{A}$-distance term should also be weighted by $\alpha$. We, therefore, rewrite the optimization problem into the following form, where the source loss and discrimination loss are weighted together:

$$\min_{\boldsymbol{A}} \quad \alpha\left(\mathcal{L}_{S,valid}(h; \boldsymbol{w}_{\mathcal{R}}, \boldsymbol{A}) - \mathcal{L}_d(h_d; \boldsymbol{w}_{\mathcal{R}}, \boldsymbol{A})\right) \tag{16}$$
$$+ (1-\alpha)\mathcal{L}_{T,valid}(h; \boldsymbol{w}_{\mathcal{R}}, \boldsymbol{A}),$$

$$\text{s.t.} \quad \max_{h_d} \min_{h, \boldsymbol{w}_{\mathcal{R}}} \alpha\left(\mathcal{L}_{S,train}(h; \boldsymbol{w}_{\mathcal{R}}, \boldsymbol{A}) - \mathcal{L}_d(h_d; \boldsymbol{w}_{\mathcal{R}}, \boldsymbol{A})\right) \tag{17}$$
$$+ (1-\alpha)\mathcal{L}_{T,train}(h; \boldsymbol{w}_{\mathcal{R}}, \boldsymbol{A}).$$

Comparing to the origin bi-level optimization in Eqs. 2 and 3, AdaptNAS introduces an adversarial loss to both levels, and the AdaptNAS-C version also introduces the target loss. This ensures the generalizability of both levels. A general difficulty in the bi-level optimization setting is the upper-level optimization highly depends on the lower-level one and is impacted by the quality of the lower-level solution. Similarly, if the solution of lower-level problem has a large generalization gap, it will be hard for the upper-level one to generalize well. The symmetrically constraint on both levels can alleviate this issue.

|  | (a) MNIST | (b) MNIST-M |
|---|---|---|
|  | (c) SVHN | |

| Search Method | Source / Target | MNIST MNIST-M | MNIST SVHN | MNIST-M SVHN |
|---|---|---|---|---|
| Search on Source | | 98.56 | 94.70 | 95.28 |
| AdaptNAS (ours) | | 98.75 | 95.63 | 95.48 |
| Search on Target | | 98.61 | 95.60 | 95.60 |

(d) Test accuracy of obtained architectures on the target domain.

Figure 1: The generalisability of AdaptNAS. Figure (a), (b) and (c) shows sample images from each domain. Table (d) shows test accuracy of obtained architectures on the target domain. The first row corresponds to ProxyNAS method without generalization constraint. The last row is our aiming performance. The middle row is our method.

A remaining problem is that for the hybrid loss calculation on CIFAR-10 and ImageNet, we cannot directly use their labels. The reason is that to let the bound in Corollary 4 work, the hypothesis $h$ should be identical for both domains. In practice, the classifier depends on the dimension of output, and there is a large gap between categories in CIFAR-10 and ImageNet (10 versus 1,000 different classes). To bridge this gap, we apply self-supervised learning. In self-supervised learning, samples are transformed and labeled based on some predefined rules. The labels are no longer correlated to objects in samples but the rule we defined to transform the samples. Besides, self-supervised learning has been demonstrated to learn feature mapping on one dataset and then well apply the learned mapping to another dataset [10, 12]. To be specific, we utilize a rotation task [10] for its impressive performance. In the rotation task, each sample in the dataset is rotated to different degrees and labeled with them. We use four different degrees: $0°$, $90°$, $180°$ and $270°$. We can therefore learn a 4-class classification task with the identical categories on both CIFAR-10 and ImageNet.

## 5 Experiments

We perform extensive experiments on various domains to demonstrate the practical generalisability of AdaptNAS. Firstly, we use three relatively small digits datasets to compare our result with the results of searching on the source domain only and on the target domain directly. Then, we search with both versions of our method under the standard NAS setting (i.e. with CIFAR-10 as the source domain and ImageNet as the target domain) for multiple times with various hyperparameters to justify our claims following Theorem 2 and 4. Finally, the obtained architectures with our optimal settings are compared with the current state-of-the-arts.

### 5.1 Search Setting

Following many previous works [6, 7, 17, 25, 26], we use the NASNet search space [26]. There are 2 kinds of cells, including normal cells and reduction cells, and each cell has 7 nodes, including 2 input nodes, 1 output node and 4 computation nodes. We use a set of 8 different candidate operations. The source dataset, CIFAR-10, contains 50,000 samples in the training set. For target domain samples, we construct a subset of 50,000 samples from ImageNet, containing 50 samples from each category, as target samples that we have access to during searching. This is about 3.90% of the entire ImageNet. More details of our experiments settings are available in the supplementary material.

### 5.2 Cross-Domain Generalization with AdaptNAS

We use three pairs of source and target domain of digits to demonstrate the generalisability of AdaptNAS. The first pair is MNIST [15] and MNIST-M [9]. MNIST is a dataset of greyscale handwritten digits (Figure 1(a)). MNIST-M modifies MNIST by blending greyscale images over random patches of colour photos in BSDS500 [1] (Figure 1(b)). The blending introduces extra colour and texture. In the second pair, we still use MNIST as the source domain, and the target domain is SVHN [18], which includes natural images of street house numbers (Figure 1(c)). The last pair still includes SVHN as the target domain, but the source domain is the more divergent MNIST-M. Intuitively, the first setting is the simplest, the second one is the hardest, and the last one is moderate.

Figure 1(d) shows the test accuracy of all obtained architectures on the target domain. Our method can consistently outperform the source only search method. It is also competitive to directly searching on the target domain and can even occasionally outperform it. This is because we only remain architectures after search and retrain the network weights from scratch. It is possible for an architecture whose generalisability is explicitly optimized to outperform another architecture searched on a single domain.

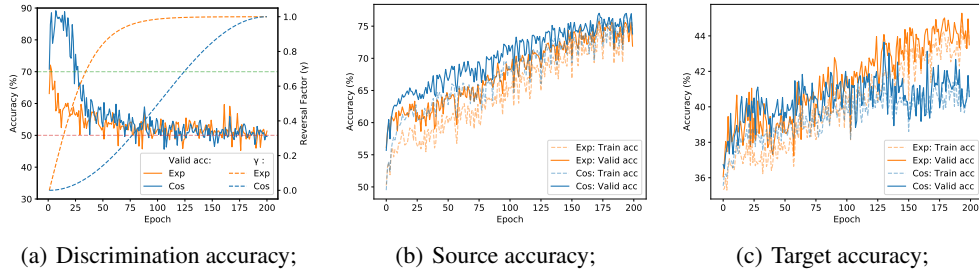

| (a) Discrimination accuracy; | (b) Source accuracy; | (c) Target accuracy; |

Figure 2: search curves

## 5.3 Better Generalization with The Hybrid Loss

Firstly, we test different parameters for the AdaptNAS-C, including $\alpha$ and $\beta$. We use 5 different values for $\alpha$ from 0 to 1 with an interval of 0.25. When $\alpha = 1$ and only the source loss is considered, AdaptNAS-C becomes AdaptNAS-S, which is identical to the first row of Table 2. A relatively new case is when $\alpha = 0$, and only the target loss is considered. We also use 3 different values for $\beta$, including 0.50, 0.83 and 0.98. Table 1 shows the validation error during search and the test error of retraining on CIFAR-10 and the full ImageNet. In the first group where $\beta = 0.50$, the source and target domain has the same number of samples (50,000 samples from each domain). With the extreme setting that $\alpha = 0$, the performance on CIFAR-10 is the worst. The target error is same to the one with $\alpha = 0.50$ but is lower then the one with $\alpha = 0.25$. Although a decent performance might be achieved by solely using target loss if there are sufficient target samples, if there are increasingly few target samples (e.g. the second group where $\beta = 0.83$ and the third group where $\beta = 0.98$), the effect of using domain discriminator loss can be even more remarkable. In the second group, the number of target samples is decreased to 10,000, and in the third group, the number is further decreased to 1,000. With less target samples, using the hybrid loss can improve the target domain performance by up to 4.4%.

Table 1: Performance of various AdaptNAS-C settings.

| $\alpha$ | $\beta$ | Source Err. (%) (CIFAR-10) | | Target Err. (%) (ImageNet) | |
|---|---|---|---|---|---|
| | | Valid | Test | Valid | Test |
| 0.00 | 0.50 | 49.26 | 3.00 | 42.52 | 24.5 |
| 0.25 | 0.50 | 30.00 | 2.97 | 40.13 | **24.2** |
| 0.50 | 0.50 | 25.16 | **2.50** | 40.13 | 24.5 |
| 0.75 | 0.50 | 22.78 | 2.62 | 42.41 | 25.1 |
| 1.00 | 0.50 | 23.15 | 2.53 | 55.37 | 25.4 |
| 0.00 | 0.83 | 52.19 | 3.21 | 53.65 | 25.5 |
| 0.25 | 0.83 | 38.06 | 3.17 | 51.56 | 25.0 |
| 0.50 | 0.83 | 33.82 | **2.95** | 49.86 | **24.7** |
| 0.75 | 0.83 | 28.68 | 3.00 | 54.17 | 25.5 |
| 1.00 | 0.83 | 23.89 | 2.98 | 56.39 | 25.8 |
| 0.00 | 0.98 | 74.80 | 3.91 | 69.65 | 29.5 |
| 0.25 | 0.98 | 67.31 | 3.66 | 70.90 | 26.5 |
| 0.50 | 0.98 | 51.93 | 3.56 | 64.25 | 25.8 |
| 0.75 | 0.98 | 40.68 | 3.02 | 62.75 | **25.1** |
| 1.00 | 0.98 | 30.15 | **2.93** | 61.85 | 25.7 |

We further compare AdaptNAS-S and C. When we introduce the target loss to AdaptNAS-C, we symmetrically introduce it to training and searching phase, because the upper-level optimization highly depends on the lower-level one. Despite that, we explore one more setting, where the target loss is solely introduce to the searching phase. Table 2 shows the test error of retraining. By using hybrid loss, the target error of architectures decreases. Even the hybrid loss is only used in searching, the improvement in target domain is remarkable. By using hybrid loss in both training and searching phase, the lowest target error is reached.

Table 2: Compare different versions of AdaptNAS.

| Hybrid Loss | | Source Err. (%) (CIFAR-10) | Target Err. (%) (ImageNet) |
|---|---|---|---|
| Train | Search | | |
| N | N | 2.77 | 25.3 |
| N | Y | 2.84 | 24.8 |
| Y | Y | 2.50 | 24.5 |

## 5.4 Gradient Reversal Scheduler in Adversarial Learning

We compare two different schedulers for $\gamma$ in Eqs. (14) and (15). An exponential scheduler is proposed by Ganin et al.[8], which updates $\gamma$ by:

$$\gamma_p = \frac{2}{1 + \exp(-10 \cdot p)} - 1, \tag{18}$$

where $p \in [0, 1]$ is the training procedure calculated by dividing the current epoch by the total number of epoch. However, as shown by the blue dashed line in Figure 2(a), the exponential scheduler rises too fast. We also test a cosine-based scheduler, which rises slower:

$$\gamma_p = \frac{1 - \cos(p \cdot \pi)}{2} \tag{19}$$

Table 4: Comparison with state-of-the-art NAS methods searching on different domain. For error rates on CIFAR-10, if a paper provides results with cutout, we use that version, because cutout always yield their best performance, and we use it too. On ImageNet, cutout is normally not used.

| Domain | Method | GPU Days | CIFAR-10 | | ImageNet | | | |
|---|---|---|---|---|---|---|---|---|
| | | | Params (M) | Err. (%) | Params (M) | +× (M) | Err. (%) | |
| | | | | | | | Top-1 | Top-5 |
| CIFAR-10 | NASNet-A [26] | 2K | 3.3 | 2.65 | 5.3 | 564 | 26.0 | 8.4 |
| | ENAS (micro) [19] | 0.45 | 4.6 | 2.89 | - | - | - | - |
| | DARTS (2nd order) [17] | 1 | 3.3 | 2.76±0.09 | 4.7 | 574 | 26.7 | 8.7 |
| | GDAS [7] | 0.21 | 3.4 | 2.93 | 5.3 | 581 | 26.0 | 8.5 |
| | P-DARTS [6] | 0.3 | 3.4 | 2.50 | 4.9 | 557 | 24.4 | 7.4 |
| | Proxyless-G [5] | 4.0 | 5.7 | 2.08 | - | - | - | - |
| | HM-NAS (2nd order) [23] | 1.4 | 1.8 | 2.41±0.05 | - | - | - | - |
| | MdeNAS [25] | 0.16 | 3.6 | 2.55 | 6.1 | ≤600 | 25.5 | 7.9 |
| CIFAR-100 | P-DARTS [6] | 0.3 | 3.6 | 2.62 | 5.1 | 577 | 24.7 | 7.5 |
| ImageNet[1] | MnasNet-A3 [21] | 3.8K[2] | - | - | 5.2 | 403 | 23.3 | 6.7 |
| | FBNet-C[3] [22] | 9 | - | - | 5.5 | 375 | 25.1 | - |
| | Proxyless-R (Mobile) [5] | 8.3 | - | - | - | - | 25.4 | 7.8 |
| | Proxyless (GPU) [5] | 8.3 | - | - | 7.1 | 465 | 24.9 | 7.5 |
| | HM-NAS[3] [23] | ~5 | - | - | 3.6 | 482 | 26.2 | - |
| | MdeNAS (CPU)[4] [25] | 2 | - | - | - | ≤600 | 24.8 | - |
| | MdeNAS (GPU)[4] [25] | 2 | - | - | - | ≤600 | 25.9 | - |
| Cross-Domain | AdaptNAS-S (Rot-1) | 0.5 | 3.6 | 2.59 | 5.2 | 575 | 24.7 | 7.6 |
| | AdaptNAS-S (Rot-4) | 1.8 | 3.5 | 2.77 | 5.0 | 552 | 25.3 | 7.8 |
| | AdaptNAS-C (Rot-1) | 0.7 | 3.9 | 2.72 | 5.4 | 603 | 24.3 | 7.4 |
| | AdaptNAS-C (Rot-4) | 2.0 | 3.7 | 2.50 | 5.3 | 583 | 24.2 | 7.4 |

[1] Include methods using subset of ImageNet.
[2] MnasNet takes 4.5 days on 64 TPUv2 for one search. The GPU days is estimated by [22].
[3] FBNet and HM-NAS searches on a subset of ImageNet with 100 classes.
[4] MdeNas searches with the MobileNetV2 [11] as backbone and accelerated by the structure in [5].

where the definition of $p$ is the same as above. Both schedulers are experimented with AdaptNAS-S, which does not consider the target domain loss, to emphasize the impact of the discriminator.

Figure 2 shows accuracy curves during search. We also retrain obtained architectures on CIFAR-10 and the entire ImageNet after search (Table 3). As shown in Figure 2(a), the initial accuracy of both discriminators is similar, and the one with an exponential scheduler immediately drops, while the one with cosine scheduler can achieve relatively high accuracy, and then declines as $\gamma$ increases. Common sense is a strong

Table 3: Test error of searching with different $\gamma$ schedulers.

| Scheduler | Source Err. (%) (CIFAR-10) | Target Err. (%) (ImageNet) |
|---|---|---|
| Exponential | 2.93 | 25.1 |
| Cosine | 2.77 | 25.3 |

discriminator usually leads to small loss and vanishing gradients [2], which makes the network hard to learn. This is verified by Figure 2(b) and 2(c). Although the cosine scheduler corresponds to a better performance in the source domain, its target domain performance is overtaken by the exponential scheduler, which means the network trained with exponential generalized better. In Table 3, the test performance by retraining also shows the same conclusion. When the cosine scheduler is used, the error is low on CIFAR-10 but is high on ImageNet which indicates the architecture is not adapted successfully.

## 5.5 Comparison with State-of-the-arts

Table 4 compares AdaptNAS with current state-of-the-art NAS methods, including methods both searching with proxy tasks or directly searching on ImageNet. We notice one drawback of using self-supervised learning is it dramatically increases the searching time. In the rotation task, where we rotate an image to 4 different degrees, the sample number increases 4 times. To balance the performance and efficiency trade-off, we add a simplified Rot-1 setting, where each sample is randomly rotated to only 1 of 4 degrees in each epoch. The origin task is then notated as Rot-4.

Comparing to methods searching on CIFAR-10, our method can reach lower ImageNet top-1 error and competitive CIFAR-10 error. The simplified AdaptNAS-S Rot-1 is our fastest setting and is even faster than several differentiable NAS methods on CIFAR-10 including DARTS, HM-NAS, and Proxyless-G. Our best top-1 error on ImageNet is reached by AdaptNAS-C with Rot-4, which costs 2 GPU days for searching. Even it is slower than many ProxyNAS methods, it is faster than most

NAS methods that directly search on ImageNet, including the ones using a subset of ImageNet. Only MdeNAS, which searches with acceleration, can reach a similar search cost.

## 6  Conclusion

In this paper, the generalization issue in ProxyNAS is studied, and two versions of generalization bound are proposed. Motivated by the generalization bound, we design a AdaptNAS method to find architectures with better generalizability. We provide a new perspective in NAS: instead of direct searching on ImageNet or its subset, optimizing the generalizability of architectures by adding domain distance constraint during the search can reach better performance with lower computation cost. Extensive experiments on CIFAR-10 and ImageNet demonstrate that AdaptNAS is a more affordable searching method with more controllable generalizability comparing to the current state-of-the-art proxy or proxyless NAS methods.

## Broader Impact

This paper provides a novel perspective of cross-domain generalization in neural architecture search towards the efficient design of neural architectures with strong generalizability. This will lead to a better understanding of the generalizability of neural architectures. The proposed method will be used to design neural architectures for computer vision tasks with affordable computation cost.

## Acknowledgement

The authors would like to thank the Area Chair and the reviewers for their constructive comments. This work was supported by the Australian Research Council under Project DE180101438.

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
