[Supplementary Material]

# Adapting Neural Architectures Between Domains
# (Supplementary Material)

**Yanxi Li** [1], **Zhaohui Yang** [2,3], **Yunhe Wang** [2], **Chang Xu** [1]
[1] School of Computer Science, University of Sydney, Australia
[2] Noah's Ark Lab, Huawei Technologies, China
[3] Key Lab of Machine Perception (MOE), Department of Machine Intelligence,
Peking University, China
yali0722@uni.sydney.edu.au, zhaohuiyang@pku.edu.cn,
yunhe.wang@huawei.com, c.xu@sydney.edu.au

This supplementary material consists of three parts, including the proofs of all lemmas, theorems and corollaries (Section A), details of the experiment setting (Section B) and some additional experiment results (Section C).

## A  Proofs

### A.1  Proof of Lemma 1

**Lemma 1.** [2] *Let $\mathcal{R}$ be a representation function $\mathcal{R} : \mathcal{X} \to \mathcal{Z}$, and $\widetilde{\mathcal{D}}_S$ and $\widetilde{\mathcal{D}}_T$ be the source and target distribution over $\mathcal{Z}$, respectively. For $h \in \mathcal{H}$:*

$$\varepsilon_T(h) \le \varepsilon_S(h) + \frac{1}{2} d_{\mathcal{H}\Delta\mathcal{H}}(\widetilde{\mathcal{D}}_S, \widetilde{\mathcal{D}}_T) + \lambda, \tag{1}$$

*where $\lambda$ is combined error of the optimum hypothesis $h^* = \arg\min_{h\in\mathcal{H}} \varepsilon_S(h) + \varepsilon_T(h)$ on both domains: $\lambda = \varepsilon_S(h^*) + \varepsilon_T(h^*)$.*

*Proof.* The proof bases on the triangle inequality for classification error [1, 4]: for any labeling function $f_1$, $f_2$ and $f_3$, $\varepsilon(f_1, f_2) \le \varepsilon(f_1, f_3) + \varepsilon(f_2, f_3)$. With the definition that $\varepsilon_S(h) := \varepsilon_S(h, \tilde{f}_S)$, for the source domain, we have:

$$\varepsilon_S(h, h^*) \le \varepsilon_S(h, \tilde{f}_S) + \varepsilon_S(h^*, \tilde{f}_S) = \varepsilon_S(h) + \varepsilon_S(h^*). \tag{2}$$

For the target domain, we use a slightly different version:

$$\varepsilon_T(h) = \varepsilon_T(h, \tilde{f}_T) \le \varepsilon_T(h^*, \tilde{f}_T) + \varepsilon_T(h, h^*) = \varepsilon_T(h^*) + \varepsilon_T(h, h^*). \tag{3}$$

The symmetric difference $\mathcal{A}$-distance has the following property:

$$\forall h, h' \in \mathcal{H}, \quad |\varepsilon_S(h, h') - \varepsilon_T(h, h')| \le \frac{1}{2} d_{\mathcal{H}\Delta\mathcal{H}}(\mathcal{D}_S, \mathcal{D}_T). \tag{4}$$

We can have:

$$
\begin{aligned}
\varepsilon_T(h) &\le \varepsilon_T(h^*) + \varepsilon_T(h, h^*) && \text{(Applying Eq. 3)}\\
&\le \varepsilon_T(h^*) + \varepsilon_S(h, h^*) + |\varepsilon_S(h, h') - \varepsilon_T(h, h')|\\
&\le \varepsilon_T(h^*) + \varepsilon_S(h, h^*) + \frac{1}{2} d_{\mathcal{H}\Delta\mathcal{H}}(\mathcal{D}_S, \mathcal{D}_T) && \text{(Applying Eq. 4)}\\
&\le \varepsilon_T(h^*) + \varepsilon_S(h) + \varepsilon_S(h^*) + \frac{1}{2} d_{\mathcal{H}\Delta\mathcal{H}}(\mathcal{D}_S, \mathcal{D}_T) && \text{(Applying Eq. 2)}\\
&= \varepsilon_S(h) + \frac{1}{2} d_{\mathcal{H}\Delta\mathcal{H}}(\mathcal{D}_S, \mathcal{D}_T) + \lambda,
\end{aligned}
$$

where $\lambda = \varepsilon_S(h^*) + \varepsilon_T(h^*)$. □

## A.2 Proof of Theorem 2

**Theorem 2.** *Let $m$ be the size of $\widetilde{\mathcal{U}}_{S,valid}$, $d'$ be the VC-dimension of $\mathcal{H}'$, and $\widetilde{\mathcal{U}}_S$ and $\widetilde{\mathcal{U}}_T$ be sets of unlabelled i.i.d. samples drawn from $\widetilde{\mathcal{D}}_S$ and $\widetilde{\mathcal{D}}_T$, each with size $m'$. With probability at least $1 - \delta$, for $h \in \mathcal{H}'$:*

$$\varepsilon_T(h) \leq \hat{\varepsilon}_{S,valid}(h) + \frac{d' \log m - \log \delta}{3m} + \sqrt{\frac{2(d' \log m - \log \delta)}{m}} \\ + \frac{1}{2} d_{\mathcal{H} \Delta \mathcal{H}}(\widetilde{\mathcal{U}}_S, \widetilde{\mathcal{U}}_T) + 4\sqrt{\frac{d' \log(2m') + \log(4/\delta)}{m'}} + \lambda. \tag{5}$$

*Proof.* Firstly, we derive the bound between the expected source error $\varepsilon_S(h)$ in Eq. 1 and the empirical source validation error $\hat{\varepsilon}_{S,valid}(h)$. Let $\kappa_i(h) = \varepsilon_S(h) - \ell(h(z_i), y_i)$ for $h \in \mathcal{H}'$ and $z_i \in \widetilde{\mathcal{U}}_{S,valid}$. Therefore,

$$\varepsilon_S(h) - \hat{\varepsilon}_{S,valid}(h) = \frac{1}{m} \sum_{i=1}^{m} \kappa_i(h). \tag{6}$$

Because $\varepsilon_S(h) \in [0,1]$ and $\ell(h(z_i), y_i) \in [0,1]$, we have $\varepsilon_S(h) - \ell(h(z_i), y_i) \in [-1,1]$ and $\mathbb{E}[\kappa_i(h)^2] \leq 1$, $|\kappa_i(h)| \leq 1$. By applying Bernstein inequality,

$$\mathbb{P}\left(\frac{1}{m} \sum_{i=1}^{m} \kappa_i(h) > \xi\right) \leq \exp\left(-\frac{\xi^2 m/2}{1 + \xi/3}\right). \tag{7}$$

By taking union bound of Eq. 7 over all $h \in \mathcal{H}'$ with VC-dimension $d'$,

$$\mathbb{P}\left(\cup_{h \in \mathcal{H}'} \frac{1}{m} \sum_{i=1}^{m} \kappa_i(h) > \xi\right) \leq m^{d'} \exp\left(-\frac{\xi^2 m/2}{1 + \xi/3}\right). \tag{8}$$

Let $\delta = m^{d'} \exp\left(-\frac{\xi^2 m/2}{1+\xi/3}\right)$ and solve the equation for $\xi$:

$$\xi = \frac{d' \log m - \log \delta}{3m} \pm \sqrt{\left(\frac{d' \log m - \log \delta}{3m}\right)^2 + \frac{2(d' \log m - \log \delta)}{m}}. \tag{9}$$

Because $\xi \geq 0$ and $\sqrt{a+b} \leq \sqrt{a} + \sqrt{b}$, Eq. 9 can be simplified as:

$$\xi \leq \frac{d' \log m - \log \delta}{3m} + \sqrt{\frac{2(d' \log m - \log \delta)}{m}}. \tag{10}$$

Thus, for any $\delta > 0$, with probability at least $1 - \delta$, for $h \in \mathcal{H}'$,

$$\varepsilon_S(h) - \hat{\varepsilon}_{S,valid}(h) \leq \frac{d' \log m - \log \delta}{3m} + \sqrt{\frac{2(d' \log m - \log \delta)}{m}}. \tag{11}$$

Finally, by applying the bound between the expected domain distance with the empirical domain distance according to [6], we can have Eq. 5. $\qquad \square$

## A.3 Proof of Lemma 3

**Lemma 3.** *Let $\varepsilon_\alpha(h)$ be an expected hybrid error weighted by $\alpha \in [0,1]$. For $h \in \mathcal{H}$:*

$$\varepsilon_T(h) \leq \varepsilon_\alpha(h) + \alpha \left(\frac{1}{2} d_{\mathcal{H} \Delta \mathcal{H}}(\widetilde{D}_S, \widetilde{D}_T) + \lambda\right). \tag{12}$$

*Proof.* According to the triangle inequality for classification error,

$$\varepsilon_T(h) \leq \varepsilon_T(h, h^*) + \varepsilon_T(h^*) \Rightarrow \varepsilon_T(h) - \varepsilon_T(h, h^*) \leq \varepsilon_T(h^*). \tag{13}$$

Similarly, for the source domain, we have

$$\varepsilon_S(h) - \varepsilon_S(h, h^*) \leq \varepsilon_S(h^*). \tag{14}$$

Therefore, the bound between the expected target error $\varepsilon_T(h)$ and the expected hybrid error $\varepsilon_\alpha(h)$ can be derived by:

$$
\begin{aligned}
|\varepsilon_T(h) - \varepsilon_\alpha(h)| &= |\varepsilon_T(h) - \alpha\varepsilon_S(h) - (1-\alpha)\varepsilon_T(h)| \\
&= \alpha|\varepsilon_T(h) - \varepsilon_S(h)| \\
&= \alpha|(\varepsilon_T(h) + \varepsilon_T(h, h^*) - \varepsilon_T(h, h^*)) - (\varepsilon_S(h) + \varepsilon_S(h, h^*) - \varepsilon_S(h, h^*))| \\
&\leq \alpha\,|(\varepsilon_T(h) - \varepsilon_T(h, h^*)) + (\varepsilon_T(h, h^*) - \varepsilon_S(h, h^*)) + (\varepsilon_S(h, h^*) - \varepsilon_S(h))| \\
&\leq \alpha\,(\varepsilon_T(h^*) + |\varepsilon_T(h, h^*) - \varepsilon_S(h, h^*)| + \varepsilon_S(h^*)) \quad \text{(Applying Eqs. 13 and 14)} \\
&\leq \alpha\left(\frac{1}{2}d_{\mathcal{H}\Delta\mathcal{H}}(\widetilde{D}_S, \widetilde{D}_T) + \lambda\right), \quad\quad\quad\quad \text{(Applying Eq. 4)}
\end{aligned}
$$

where $\lambda = \varepsilon_S(h^*) + \varepsilon_T(h^*)$. $\qquad\qquad\square$

### A.4 Proof of Corollary 4

**Corollary 4.** *Let $\alpha \in [0,1]$ be the weight of the hybrid error, and $\beta \in [0,1]$ be the ratio of i.i.d. samples drawn from $\widetilde{\mathcal{D}}_S$ and $\widetilde{\mathcal{D}}_T$ in a held-out validation set. With probability at least $1-\delta$, for $h \in \mathcal{H}'$:*

$$
\begin{aligned}
\varepsilon_T(h) \leq\; &\hat{\varepsilon}_{\alpha,valid}(h) + \left(\frac{\alpha}{\beta} + \frac{1-\alpha}{1-\beta}\right)\left(\frac{d'\log m - \log\delta}{3m} + \sqrt{\frac{2(d'\log m - \log\delta)}{m}}\right) \\
&+ \alpha\left(\frac{1}{2}d_{\mathcal{H}\Delta\mathcal{H}}(\widetilde{\mathcal{U}}_S, \widetilde{\mathcal{U}}_T) + 4\sqrt{\frac{d'\log(2m') + \log(4/\delta)}{m'}} + \lambda\right).
\end{aligned}
\tag{15}
$$

*Proof.* By combining Theorem 2 and Lemma 3, we can derive the proof of Corollary 4. Let $\widetilde{\mathcal{U}}_{\beta,valid}$ be a hybrid validation set with $\boldsymbol{z}_i \in \widetilde{\mathcal{U}}_{\beta,valid}$ for $i \in [1, \beta m]$ from source domain and $\boldsymbol{z}_i \in \widetilde{\mathcal{U}}_{\beta,valid}$ for $i \in [\beta m + 1, m]$ from target domain. Let $\kappa_i(h) = (\alpha/\beta)(\varepsilon_S(h) - \ell(h(\boldsymbol{z}_i), y_i))$ for $i \in [1, \beta m]$, and $\kappa_i(h) = (1-\alpha)/(1-\beta)(\varepsilon_S(h) - \ell(h(\boldsymbol{z}_i), y_i))$ for $i \in [\beta m + 1, m]$. Therefore,

$$
\begin{aligned}
&\varepsilon_{\alpha,valid}(h) - \hat{\varepsilon}_{\alpha,valid}(h) \\
=\;&\alpha(\varepsilon_{S,valid}(h) - \hat{\varepsilon}_{S,valid}(h)) + (1-\alpha)(\varepsilon_{T,valid}(h) - \hat{\varepsilon}_{T,valid}(h)) \\
=\;&\frac{\alpha}{\beta m}\sum_{i=1}^{\beta m}(\varepsilon_S(h) - \ell(h(\boldsymbol{z}_i), y_i)) + \frac{1-\alpha}{(1-\beta)m}\sum_{i=\beta m+1}^{m}(\varepsilon_S(h) - \ell(h(\boldsymbol{z}_i), y_i)) \\
=\;&\frac{1}{m}\sum_{i=1}^{m}\kappa_i(h).
\end{aligned}
\tag{16}
$$

The rest of the proof is similar to the proof of Theorem 2, but with $\mathbb{E}[\kappa_i(h)^2] \leq (\alpha/\beta + (1-\alpha)/(1-\beta))^2$ and $|\kappa_i(h)| \leq \alpha/\beta + (1-\alpha)/(1-\beta)$. We can have: for any $\delta > 0$, with probability at least $1-\delta$, for $h \in \mathcal{H}'$,

$$
\varepsilon_{\alpha,valid}(h) - \hat{\varepsilon}_{\alpha,valid}(h) \leq \left(\frac{\alpha}{\beta} + \frac{1-\alpha}{1-\beta}\right)\left(\frac{d'\log m - \log\delta}{3m} + \sqrt{\frac{2(d'\log m - \log\delta)}{m}}\right)
\tag{17}
$$

Finally, by applying the bound between the expected domain distance with the empirical domain distance according to [6], we can have Eq. 15. $\qquad\square$

## B Experiment Details

### B.1 NAS Search Space

Following many previous works [3, 5, 7, 9, 10], we use the NASNet search space [10]. There are 2 kinds of cells in the search space, including normal cells and reduction cells. Normal cells use stride 1 and maintain the size of feature maps. Reduction cells use stride 2 and reduce the height and width of feature maps to a half. After a reduction cell, the channel number is doubled. Each cell has 7 nodes, including 2 input nodes, 1 output node and 4 computation nodes. The connection pattern of

cells in the NASNet search space is illustrated in Figure 1, where $h_{i-2}$ and $h_{i-1}$ are input nodes connected to the previous two cells, $h_i$ is an output node concatenating all computation nodes of the current cell, and $x^{(0)}$ to $x^{(3)}$ are computation nodes taking outputs of previous nodes as their inputs and applying some operations on them. Cells are stacked sequentially to build a network. In the network, cells located at the $1/3$ and $2/3$ are reduction cells, while others are normal cells.

Figure 1: The connection pattern of cells in the NASNet search space.

We use a set of 8 different candidate operations, including:

- $3 \times 3$ separable convolution;
- $5 \times 5$ separable convolution;
- $3 \times 3$ dilated separable convolution;
- $5 \times 5$ dilated separable convolution;
- $3 \times 3$ max pooling;
- $3 \times 3$ average pooling;
- identity (i.e. skip-connection);
- zero (i.e. not connected).

All the operations follow the ReLU-Conv/Pooling-BN pattern except identity and zero.

## B.2 Search and Evaluation on Digits

For searching on digits datasets, we use a network with 5 cells, where the 2nd and 3rd cells are reduction cells. The first cell has 16 initial channels. We search for 100 epochs. After searching, we use the same network size for evaluation. The network is trained for 100 epochs.

## B.3 Search and Evaluation on CIFAR-10 and ImageNet

For searching on CIFAR-10 and ImageNet, we use a network with 8 cells, where the 3rd and 6th cells are reduction cell. The first cell has 16 initial channels. We search for 200 epochs. For evaluation on CIFAR-10, we use a network with 20 cells and 36 initial channels. The network is trained for 600 epochs with cutout. The network for ImageNet generalization is relatively shallow but wide, which has 14 cells and 48 initial channels. The network is trained for 250 epochs. Auxiliary heads are used for evaluation on both datasets, which is inserted after the 2nd reduction cell. We follow DARTS [7] and PC-DARTS [8] and train networks for 600 and 250 epochs on CIFAR-10 and ImageNet, respectively.

# C More Results

## C.1 Latent Space Visualization (on Digits)

We visualize the latent space learned during search. The setting with MNIST as the source domain and SVHN as the target domain, which has the maximum generalization gap (5.3% test error by searching in the source domain versus 4.4% test error by searching in the target domain), is selected for demonstration. 500 random samples from each domain are chosen. Figure 2 shows the latent space learned by different searching methods, including searching in the source domain only and searching with AdaptNAS-S. The origin latent representation is 256-dimension and is reduced to 2-dimension with t-distributed Stochastic Neighbor Embedding (t-SNE). Figure 2(a) shows the latent space learned by searching in the source domain only. As can be seen, samples from MNIST clusters by their labels, while samples from SVHN distributes almost randomly. When they are mixed together, there are more than one cluster for each label. Figure 2(b) shows the latent space learned by AdaptNAS-S. Samples from both domains clusters well. When they are mixed together, there are only one major cluster for each label with several outliers.

## C.2 Architectures of Reported Results (on CIFAR-10 and ImageNet)

Figure 3 shows architectures of the reported results compared with SOTAs in the paper. Both normal and reduction cells found by different AdaptNAS settings are provided.

(a) Search in the source domain;

(b) Search with AdaptNAS-S.

Figure 2: Latent spaces learned by different searching methods with MNIST as the source domain and SVHN as the target domain. The dimension is reduced with t-SNE. Different colors stand for different categories. There are 10 categories for different digits from 0 to 9.

(a) Normal: AdaptNAS-S (Rot-1);

(b) Reduction: AdaptNAS-S (Rot-1);

(c) Normal: AdaptNAS-S (Rot-4);

(d) Reduction: AdaptNAS-S (Rot-4);

(e) Normal: AdaptNAS-C (Rot-1);

(f) Reduction: AdaptNAS-C (Rot-1);

(g) Normal: AdaptNAS-C (Rot-4);

(h) Reduction: AdaptNAS-C (Rot-4).

Figure 3: Architectures found by different settings of AdaptNAS