[Reviews · NeurIPS 2020]

Review 1

Summary and Contributions: This work aims to minimize the cross-domain generalization gap that generally exists in current neural architecture search (NAS) methods with proxy tasks. Instead of directly using the target dataset for searching, which suffers from the high computation cost, the authors propose to improve the generalizability of neural architectures by leveraging a small portion of target samples via a domain adaptation technique. They first theoretically analyze the generalization bounds of architectures in NAS methods with proxy tasks, and then design a novel approach based on the analysis to minimize the bounds via adapting architectures between domains. Overall, the theoretical analysis is solid, and the experiments show that the proposed approach can achieve a good trade-off between the search cost and the target domain accuracy.

Strengths: There are several strengths of this work: - The motivation of this work is clear. The generalization issue between CIFAR-10 and ImageNet is a general problem in NAS. Existing methods aim to solve this problem by direct searching on ImageNet, which increases the search cost dramatically. This work provides a novel perspective to solve this problem. - The theoretical analysis is very solid and reveals important facts of how the generalization gap of neural architectures can be bounded. This can be used as a guideline for the algorithm design. - The proposed approach is based on and consistent with the theoretical results. Both the algorithm and theory are well supported by experiments. - Experiments of this work are very solid, including different pairs of domains and detailed ablation study, which makes this work convincing.

Weaknesses: Besides the explicit constraint on domain distance, which is the main contribution of the proposed method, there is another difference from this work to the others, that is the self-supervised technique which is used to bridge the inconsistency of labels between domains. This is rarely discussed in the paper. The authors should discuss what the role of self-supervised in this work is and to what extent does it contributes to the final result.

Correctness: The claims, method and the empirical methodology are correct.

Clarity: This paper is well written.

Relation to Prior Work: Yes

Reproducibility: Yes

Additional Feedback: There are several questions to the authors: - In the ablation study part, both the validation and test error rates are reported in Table 1. However, only the test error rates are reported in Tables 2 and 3. Could the authors explain why there is such an inconsistency? - In FBNet and HM-NAS, the searches are on a subset of ImageNet with 100 classes, while this work uses a different manner to construct the subset of ImageNet. Please compare the two methods and explain why the latter is applied. UPDATE after rebuttal: I have read the authors' rebuttal and other reviews. The paper has an interesting idea to adapt neural architecture between domains, and the authors suggest the possibility of the AdaptNAS to be a strong plus over existing NAS methods. It is also nice to see a more detailed analysis of the hyperparameters in the rebuttal. I would like to recommend acceptance.


Review 2

Summary and Contributions: This paper aims at improving the generalization of neural architectures via domain adaptation. This paper has analyzed the generalization bounds of the derived architecture and found its close relations with the validation error and the data distribution distance on both domains. This paper has proposed AdaptNAS, a novel and principled approach to adapt neural architectures between domains in NAS.

Strengths: In this paper, the generalization issue in ProxyNAS is studied, and two versions of generalization bound are proposed. Motivated by the generalization bound, this paper has designed an AdaptNAS method to find architectures with better generalizability. This paper has provided a new perspective in NAS: instead of direct searching on ImageNet or its subset, optimizing the generalizability of architectures by adding domain distance constraint during the search can reach better performance with lower computation cost. Extensive experiments on CIFAR-10 and ImageNet demonstrate that AdaptNAS is a more affordable searching method with more controllable generalizability comparing to the current state-of-the-art proxy or proxyless NAS methods.

Weaknesses: In Tab 4, the proposed method has no obvious improvement compared to P-DARTS. Figure 1: Table (d) is not reproducible in my implementation. My implementation of AdaptNAS can not outperform 'search on target' and there is even an obvious performance gap. Please also include experiments of SVHN(source) and MNIST(target) as in this paper: https://arxiv.org/pdf/1505.07818.pdf After I read the comments from other reviewers and the rebuttal, I guess there may be some problems in my implementation and this paper is a very nice submission.

Correctness: The prove in the supp looks correct when I follow the logic of the author.

Clarity: The writing of this paper is good except for several typos.

Relation to Prior Work: NAS applied in domain adaptation is new.

Reproducibility: No

Additional Feedback: Please double check the experiments in Figure 1: Table (d) .


Review 3

Summary and Contributions: The authors investigate the generalization gap of neural architectures between two different domains. Based on the analysis of generalization ability, the authors propose an AdaptNAS method by applying domain adaptation techniques to neural architecture search (NAS). Specifically, the proposed method incorporates a domain distance constraint and cross-domain self-supervised learning technique into the training of NAS models. Extensive experiments on CIFAR-10 and ImageNet demonstrate the superiority of AdaptNAS over existing methods.

Strengths: 1. It is worth mentioning that the authors theoretically analyze the generalization ability of architectures searched by NAS between two different domains. 2. The authors propose a theory-inspired AdaptNAS method by incorporating the domain adaptation and the self-supervised learning techniques into NAS. 3. Experimental results demonstrate the effectiveness of the proposed method on two benchmark datasets.

Weaknesses: 1. Some notations are very confusing. The authors use alpha to denote both the searched architecture and a scalar hyperparameter in Eq. (17). 2. The authors claim that the proposed method is able to reduce the generalization gap of architectures between two different domains. How much gap can be reduced in practice? It would be stronger to provide more discussions and results to illustrate this. 3. The self-supervised learning task seems to be very important for the training of the proposed method. What would happen if the authors use a different self-supervised learning task (e.g., solving jigsaw puzzles [1])? More discussions should be provided. 4. What would happen if the authors use a larger subset of ImageNet as the target samples during the search? Can the proposed method use the entire ImageNet dataset? 5. More implementation details should be provided. How many epochs do the authors train the model on CIFAR-10 and ImageNet? Do the authors randomly sample images from each category to construct the subset of ImageNet? Refs [1] Noroozi, Mehdi, and Paolo Favaro. "Unsupervised learning of visual representations by solving jigsaw puzzles." European Conference on Computer Vision. Springer, Cham, 2016.

Correctness: Yes

Clarity: Yes

Relation to Prior Work: Yes

Reproducibility: Yes

Additional Feedback: NA


Review 4

Summary and Contributions: This paper addresses the problem of NAS. The main idea is to bridge the domain discrepancy between the proxy task dataset and the target dataset by domain adaptation. A theoretical generalization bound is analyzed and a corresponding algorithm is given, namely AdaptNAS.

Strengths: The motivation is sound. The idea is novel and the proofs are detailed. I like the idea of using a transformed task as a latent space for domain discriminator.

Weaknesses: 1. In AdaptNAS, a domain discriminator is used to approximate the domain discrepancy, which might introduce a certain amount of computation overhead. 2. The results only show marginal improvement compared to previous state-of-the-art, especially P-DARTS[6] and MdeNAS[23]. In particular, [23] performs consistently better than the proposed method while only searching in CIFAR-10. 3. More importantly, it seems like there is no ablation study between using L_d (domain adaptation loss) or not. This makes it difficult to identify whether the performance is caused by using training data from both domains (L_S, L_T) or by the domain adaptation loss (L_d), which is the main contribution. 4. Also, in Tab 1, when alpha=0, there is indeed no L_d. However, it performs even better than most of the other settings where L_d presents (alpha > 0). This indicates the proposed L_d is less effective than directly utilizing data from the target domain. 5. What causes the inconsistency between Rot-4 and Rot-1? For the AdaptNAS-S, the Rot-4 version performs worse, while for the AdaptNAS-C, Rot-4 version performs better with even smaller network capacity. ------------------------------------------------- After rebuttal: The author convince me with a detailed explanation of my concerns. I encourage the author add these details to the final version of the paper.

Correctness: As listed above, a key ablation study is missing.

Clarity: The paper is well written and easy to follow. A typo in L267: “deceasing”

Relation to Prior Work: Yes.

Reproducibility: Yes

Additional Feedback: I mainly give my score mainly based on point 2, 3, 4 listed in the weakness. I like the general idea of this paper, but whether the experiment can consistently validate its effectiveness is a more important criteria for publication at NeurIPS.

[Author Response · NeurIPS 2020]

We thank the reviewers for their insightful feedback and constructive advice. In the following, we address their concerns and questions.

**[R1 Self-supervision]** Self-supervision is a trick used to bridge the difference of labels between CIFAR-10 and ImageNet. We also concern the impact of it. Although the calculation of supervised loss on the blended CIFAR-10-ImageNet dataset is intractable, we perform ablation study on digits domains with supervised loss to show the effectiveness of solely using AdaptNAS.

**[R1 Valid Error]** In previous one-shot NAS works, the validation performances are not reported, because one-shot models are not trained completely during search. We follow them in other tables. However, in Table 1, we want to show the impact of hyper-parameters of the hybrid loss during search, so we report validation errors only in this specific table.

**[R1 Subset construction]** FBNet and HM-NAS use 100 out of 1000 classes, which is 10% of the ImageNet, and we use fewer samples than them (3.90%). In this case, if we only use several classes, we concern the data might lose divergence and be biased. Thus, we include 50 random samples from each of all the categories.

**[R2 Digits results]** Thanks for checking our experiment results. We double-checked our implementation and logs and found the results are correct. The test accuracy on MNIST with SVHN as the source domain by searching on source, searching with AdaptNAS and searching on target is 99.33%, 99.21% and 99.20%, respectively. The test accuracy by searching on source is the best. We deem this abnormal result is because the source SVHN is much more complicated than the target MNIST, which is opposite to the case of NAS.

**[R3 Notation]** Thanks for reminding. We will replace $\alpha$ for architecture parameter with $A$ to distinguish with $\alpha$.

**[R3 Gap in practice]** A popular quantitative metrics for domain gap is the $\mathcal{A}$-distance, which we approximate with a domain discriminator during search, but it is inappropriate to use it during evaluation, because networks on CIFAR-10 and ImageNet are trained with different scales and targets. However, we provide a visualization of feature alignment on digits dataset in Figure 2 of the supplementary material, which gives a side view of the generalization gap in practice.

**[R3 Self-supervision]** We select the Rotation task because it is a promising yet simple task and can be easily format as a typical classification task where the network takes a single augmented image as the input and predicts its rotation degree as the label. This is consistent with our theorems. However, this does not always hold in other self-supervised tasks. For example, in Jigsaw, the network takes 9 images patches as input and outputs a conditional probability density function of the spatial arrangement. In another task, Exemplar, the triplet loss is used, and explicit class labels are avoided. Considering those reasons, we selected the Rotation task.

**[R3 Larger ImageNet subset]** In theory, a larger subset or the complete ImageNet can be used for searching, but the computation cost will raise significantly. As we aim to leverage a small number of target samples to decrease the cross-domain generalization gap under limited computation resources, we set $\beta = 0.5$, where both domains have identical number of samples, as the upper bound and gradually decrease the ratio of target samples.

**[R3 Implementation Details]** We follow DARTS and PC-DARTS and train networks for 600 and 250 epochs on CIFAR-10 and ImageNet, respectively. Samples in the Imagenet subset are randomly chosen from each category. We are glad to add more details in the supplementary material.

**[R4 Computation overhead]** Computation overhead of the domain discriminator (an MLP) is very minor (1.31M FLOPs) comparing to those convolutional layers (40M - 80M FLOPs for sampled architectures) during search.

**[R2 R4 Improvement over SOTA]** Regarding MdeNAS, all of our settings consistently outperform them in terms of ImageNet top-1 error (24.7%, 25.3%, 24.3% and 24.2% comparing to 25.5% in Table 4). We used GDAS as our baseline method, because it is easy to be implemented. Being orthogonal with these related works, we can boost NAS performance from a new perspective. For example, if we adapt the architecture searched by P-DARTS with our method, we can further boost the performance to a 23.8% top-1 error on ImageNet.

**[R4 Without $L_D$]** If the number of target samples is large (e.g. $\beta = 0.5$), searching without DA loss reaches an ImageNet test error of 24.7% comparing to 24.2% reached by using DA loss. If the number of target samples reduces (e.g. $\beta = 0.83$ or $0.98$), the ImageNet test error raises to 25.5% and 27.3%, and searching with DA loss still keeps a competitive error rate of 24.7% and 25.1%. With the proposed algorithm, a few target sample can already benefit the generalization a lot, which is consistent with general DA findings.

**[R4 The $\alpha = 0$ case]** If there are sufficient target samples (e.g. $\beta = 0.5$), a decent performance might be achieved by solely using target loss (i.e. $\alpha = 0$). But if there are increasingly few target samples (e.g. $\beta = 0.83$ and $0.98$), the effect of $L_d$ can be even more remarkable, e.g. 25.5% ($\alpha = 0$) v.s. 24.7% ($\alpha = 0.5$) under $\beta = 0.83$, as shown in Tab. 1.

**[R4 Rot-4 and Rot-1]** Rot-4 rotates each sample 4 times to different directions and therefore enlarges the dataset. Rot-1 randomly selects one direction for each sample per epoch and keeps the size of dataset small. According to Table 4, AdaptNAS-S tends to choose small architectures, and such small architectures (e.g. 552M FLOPs) are difficult to fit the dataset enlarged by Rot-4. Thus, AdaptNAS-S performs well with Rot-1. In the contrary, AdaptNAS-C tends to choose large architectures (e.g. 583M FLOPs), which fits these enlarged dataset easily and yields good performance with Rot-4.

Table 1: Performance of various AdaptNAS-C settings.

| $\alpha$ | $\beta$ | Source Err. (%) (CIFAR-10) | | Target Err. (%) (ImageNet) | |
|---|---|---|---|---|---|
| | | Valid | Test | Valid | Test |
| 0.00 | 0.50 | 49.26 | 3.00 | 42.52 | 24.5 |
| 0.25 | 0.50 | 30.00 | 2.97 | 40.13 | **24.2** |
| 0.50 | 0.50 | 25.16 | **2.50** | 40.13 | 24.5 |
| 0.75 | 0.50 | 22.78 | 2.62 | 42.41 | 25.1 |
| 1.00 | 0.50 | 23.15 | 2.53 | 55.37 | 25.4 |
| 0.00 | 0.83 | 52.19 | 3.21 | 53.65 | 25.5 |
| 0.25 | 0.83 | 38.06 | 3.17 | 51.56 | 25.0 |
| 0.50 | 0.83 | 33.82 | **2.95** | 49.86 | **24.7** |
| 0.75 | 0.83 | 28.68 | 3.00 | 54.17 | 25.5 |
| 1.00 | 0.83 | 23.89 | 2.98 | 56.39 | 25.8 |
| 0.00 | 0.98 | 74.80 | 3.91 | 69.65 | 29.5 |
| 0.25 | 0.98 | 67.31 | 3.66 | 70.90 | 26.5 |
| 0.50 | 0.98 | 51.93 | 3.56 | 64.25 | 25.8 |
| 0.75 | 0.98 | 40.68 | 3.02 | 62.75 | **25.1** |
| 1.00 | 0.98 | 30.15 | **2.93** | 61.85 | 25.7 |

[Meta-Review · NeurIPS 2020]

This paper focuses on the intersection between neural architecture search and domain adaptation. The proposal is to minimize the cross-domain generalization gap that generally exists in current neural architecture search (NAS) methods with proxy tasks. The philosophy behind sounds quite interesting to me. Namely, instead of directly using the target dataset for searching, which suffers from the high computation cost, the authors propose to improve the generalizability of neural architectures by leveraging a small portion of target samples via a domain adaptation technique. This philosophy leads to a novel algorithm design I have never seen, i.e., AdaptNAS. The clarity and novelty are clearly above the bar of NeurIPS. While the reviewers had some concerns on the significance, the authors did a particularly good job in their rebuttal. Thus, all of us have agreed to accept this paper for publication! Please carefully address R4' comments in the final version.